# Unveiling the Pitfalls of Knowledge Editing for Large Language Models

**Zhoubo Li**[1,2], **Ningyu Zhang**[1,2]*, **Yunzhi Yao**[1,2], **Mengru Wang**[1,2], **Xi Chen**[4], **Huajun Chen**[1,2,3]*

[1]College of Computer Science and Technology, Zhejiang University
[2]ZJU-Ant Group Joint Research Center for Knowledge Graphs, Zhejiang University
[3]ZJU-Hangzhou Global Scientific and Technological Innovation Center, Zhejiang University [4]Tencent
{zhoubo.li,zhangningyu}@zju.edu.cn

## Abstract

As the cost associated with fine-tuning Large Language Models (LLMs) continues to rise, recent research efforts have pivoted towards developing methodologies to edit implicit knowledge embedded within LLMs. Yet, there's still a dark cloud lingering overhead – will knowledge editing trigger butterfly effect? since it is still unclear whether knowledge editing might introduce side effects that pose potential risks or not. This paper pioneers the investigation into the potential pitfalls associated with knowledge editing for LLMs. To achieve this, we introduce new benchmark datasets and propose innovative evaluation metrics. Our results underline two pivotal concerns: (1) **Knowledge Conflict**: Editing groups of facts that logically clash can magnify the inherent inconsistencies in LLMs—a facet neglected by previous methods. (2) **Knowledge Distortion**: Altering parameters with the aim of editing factual knowledge can irrevocably warp the innate knowledge structure of LLMs. Experimental results vividly demonstrate that knowledge editing might inadvertently cast a shadow of unintended consequences on LLMs, which warrant attention and efforts for future works[1].

## 1 Introduction

Despite their impressive abilities, Large Language Models (LLMs) such as ChatGPT are unaware of events occurring after their training phase and may inadvertently generate harmful or offensive content. To address this concern, the concept of **knowledge editing** for LLMs has been proposed (Cao et al., 2021; Dai et al., 2022; Mitchell et al., 2022a;b; Meng et al., 2022; 2023; Ilharco et al., 2023; Onoe et al., 2023; Zhong et al., 2023; Yao et al., 2023), which provides an efficient way to change the behavior of LLMs without resorting to an exhaustive retraining or continuous training procedure. While existing knowledge editing approaches for LLMs have demonstrated impressive results, an ancient Chinese poetic wisdom resonates a dark cloud lingering overhead: *A single hair can move the whole body.* The poetic phrase underscores the notion that even minor alterations to LLMs may lead to significant outcomes, akin to the butterfly effect observed in chaos theory (Lorenz, 2000). This raises a new critical research question: does knowledge editing for LLMs introduce irreversible unforeseen side effects?

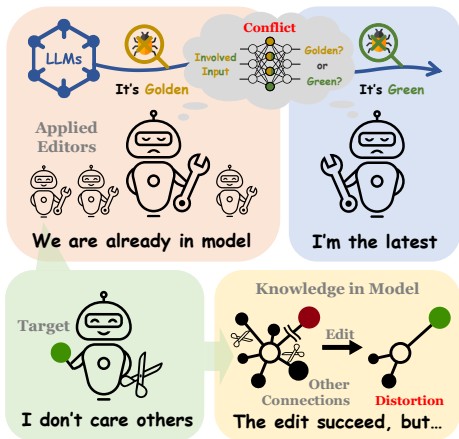

Figure 1: As the number of edits increases, the model might manifest **Knowledge Conflict** when dealing with inputs involved with multiple consecutive edits. Meanwhile, each edit could potentially lead to ruptures in knowledge links within the model, resulting in **Knowledge Distortion**.

---

*Corresponding author.

[1]Code and data are available at https://github.com/zjunlp/PitfallsKnowledgeEditing.

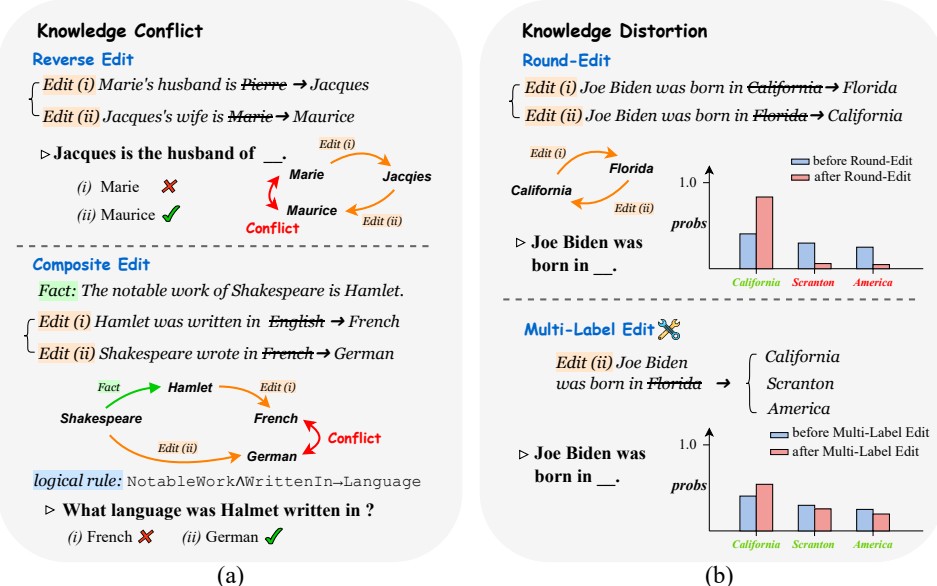

Figure 2: Unveiling the pitfalls of knowledge editing for LLMs. (a) Through REVERSE EDIT and COMPOSITE EDIT, we can observe that previous knowledge editing approaches may trigger **Knowledge Conflict**, leading to failures of knowledge editing; (b) Through ROUND-EDIT, we notice that previous knowledge editing approaches may lead to **Knowledge Distortion**, and the underlying knowledge structure within LLMs can be disrupted.

In Figure 1, we depict two types of side effects caused by knowledge editing: **Knowledge Conflict** and **Knowledge Distortion**. Using pilot experiments as an illustration, as depicted in Figure 2 (a) top, applying one single edit *(i) Marie's husband is Pierre → Jacques* to modify a fact within LLMs is straightforward using previous knowledge editing techniques. However, as the number of edits grows, these methods might encounter problems, especially when edits exhibit potential correlation. For example, consider a subsequent edit *(ii) Jacques's wife is Marie → Maurice* (REVERSE EDIT), both edit *(i)* and *(ii)* individually try to achieve their intended goals, but the model might still respond with *Marie* when prompted with *Jacques is the husband of ___*. This situation exemplifies the **Knowledge Conflict** issue. Similarly, as shown in the following composite setting (Figure 2 (a) bottom), two edits can incur a conflict due to a common logical rule (COMPOSITE EDIT), thereby leading to an inconsistency of knowledge in the model.

Further, as shown in Figure 2 (b) top, we conduct a ROUND-EDIT experiment, which means we try to restore an edited LLM (with edit *(i)*) via another subsequent edit (*(ii)*). Unfortunately, we obverse that the model after ROUND-EDIT has significantly reduced the capability to provide *Scranton* and *America* when prompting with *Where was Joe Biden born?* indicating that the implicit knowledge structure in LLMs has been disrupted. This situation exemplifies the **Knowledge Distortion** issue, which illustrates that knowledge editing may have an irreversible damage to LLMs.

Inspired by these observations, we pioneer the investigation to uncover the potential pitfalls of knowledge editing and categorize these pitfalls into two major types: knowledge conflict and knowledge distortion. To analyze the extent of these pitfalls, we create benchmark datasets tailored to prevailing methods. Specifically, we construct a dataset named CONFLICTEDIT, comprising pairs of edits that could cause knowledge conflict, and introduce a set of new metrics designed to quantify the magnitude of knowledge conflicts. We empirically observe that previous knowledge editing approaches all suffer from the knowledge conflict issue. Then, to explore the knowledge distortion issue, we create the ROUNDEDIT dataset and conduct ROUND-EDIT experiments using various knowledge editing techniques. Our findings illustrate that previous knowledge editing approaches can indeed lead to adverse consequences on the implicit knowledge structure of LLMs. We further design a simple-yet-effect method called **Multi-label Edit** (**MLE**) that combines multiple correct labels of the edit to a single process, as depicted in Figure 2 (b) bottom, which is capable to alleviate knowledge distortion and restore similar behavior compared with the original model.

## 2 EXPLORING THE PITFALLS OF KNOWLEDGE EDITING FOR LLMS

### 2.1 OVERVIEW

In this section, we briefly introduce the definition of knowledge editing and the common evaluation paradigms in previous works. Next, we outline our evaluation criterion and delve into the specifics of our analysis for knowledge editing. The representative methods used are also summarized here.

**Definition of Knowledge Editing for LLMs**  With daily updating of factual knowledge, LLMs always struggle to keep the parameters up-to-date. Knowledge editing releases the model from expensive retraining through precisely updating several outdated factual pieces of knowledge. Suppose a factual knowledge as a triplet $(s, r, o)$, an edit $e = (s, r, o \rightarrow o^*)$ modifies the object from $o$ to $o^*$ for given subject $s$ and relation $r$. After applying this edit to a language model $f_\theta$ (where $\theta$ denotes the model's parameters), there is a knowledge update applied to the model, that is

$$\begin{cases} k_o = (s, r, o) \\ k_n = (s, r, o^*) \end{cases} \tag{1}$$

where $k_o$ is the old knowledge and $k_n$ is the new one. Generally, we witness an update from $k_o$ to $k_n$ through the variation of their generation probabilities.

**Vanilla Evaluation**  The current evaluation criterion primarily focuses on various samples around an isolated edit $e = (s, r, o \rightarrow o^*)$. The post-edit model $f_{\theta'}$ (edited model) is expected to respond $o^*$ to related samples $\mathcal{I}(e)$ (excluding $e$) while still keeping the original output to unrelated samples $\mathcal{O}(e)$. Commonly, instances sampling in the neighbor of $e$ are used to build $\mathcal{I}$. Meng et al. (2022) samples $\mathcal{O}$ from a unrelated set $\{(s', r, o); s' \neq s\}$, which restricts the actuating scope of $e$. The **Reliability** metric evaluates results on the fact $(s, r, o^*)$, the **Generalization** and **Locality** metric evaluate results on $\mathcal{I}$ and $\mathcal{O}$ respectively. These metrics effectively constrain certain edits to an expected actuating scope (Meng et al., 2022; Wang et al., 2023b; Yao et al., 2023). In our experiments, for each edit $e$, we compute the results by averaging the performance of the label over the edit itself $e$ and the related samples $I(e)$.

**Motivation and Evaluation Principle**  Since LLMs can be regarded as large knowledge bases (Petroni et al., 2019), knowledge editing is expected to handle thousands of edits while maintaining knowledge coherence and integrity. In previous works, multiple edits are applied either through sequential editing (Huang et al., 2023) or mass editing (Meng et al., 2023) settings preventing the model from obvious performance drop. Unfortunately, current isolated evaluations fail to consider the interaction between the accumulating edits which could lead to subtle side effects of the model's knowledge. Hence, we start exploring the pitfalls of knowledge editing for LLMs from typical edits and focus on two new issues, namely, **Knowledge Conflict** (§2.2) and **Knowledge Distortion** (§2.3). Intuitively, The knowledge conflict setting can examine how two edits may interact or contradict each other, resulting in misinformation. On the other hand, The knowledge distortion setting can analyze potentially irreversible damage to knowledge structure with mass edits. Rather than isolated edits, these settings can comprehensively evaluate the ability of knowledge editing methods to process numerous edits without degradation, contradiction, or loss of unrelated knowledge.

**Editing Methods**  In the following Sections, we build benchmarks CONFLICTEDIT and ROUNDEDIT and conduct experiments to evaluate knowledge editing approaches as follows (we do not consider approaches that preserve parameters in LLMs):

- **Fine-tuning (FT)** updates the model's parameters by gradient descent in a certain layer through Adam and early stop strategy to maximize the probability of the editing target.
- **MEND** Mitchell et al. (2022a) utilizes a hypernetwork, which gives a low-rank update to the original fine-tuning gradients based on the edit knowledge.
- **ROME** Meng et al. (2022) uses causal tracing to locate the key layer associated with the edit knowledge, and then impose a update to the MLP module.
- **MEMIT** Meng et al. (2023) follows the locating methods in ROME and is capable of updating multiple layers at one time when editing massive knowledge.

## 2.2 KNOWLEDGE CONFLICT ANALYSIS

Note that a **robust** knowledge editing method should be supposed to deal with multiple updates of a specific fact. However, with the increasing extent of edits, there is a possibility that interference occurs between different edits, causing the former edit invalid. Particularly, when the edits are logically connected, it's challenging yet crucial to nullify previous modifications; failing to do so can lead to inconsistencies in the post-edited model. To this end, we define a notion of **Knowledge Conflict** and attempt to explore when and why the conflicts occur, and how to handle them.

### 2.2.1 PROBLEM DEFINITION

**Knowledge Conflict** Consider the scenario depicted in Figure 2(a) top, where two consecutive edits, $e_1$:*Marie's husband is Pierre → Jacques* and $e_2$:*Jacques's wife is Marie → Maurice*, are applied to a language model. These two edits both modify the fact *(Jacques, HusbandOf, ?)*. Ideally, a reliable and robust edit method should retain only the latest change. However, the current editing method may inadvertently retain knowledge from previous edits, leading to a logical inconsistency, resulting in the knowledge conflict issue. There are many factors that can contribute to the emergence of knowledge conflicts and here we consider two major scenarios: REVERSE EDIT and COMPOSITE EDIT.

**Reverse Edit** This scenario causes conflict by editing the facts with reverse relations. We formalize this situation as a pair of consecutive edits that contain reverse relations, that is

$$\text{Reverse Edit}: \begin{cases} e_1 = (s_1, r_1, o_1 \to o_2) \\ e_2 = (o_2, r_2, s_1 \to s_2) \end{cases} \quad (2)$$

where $r_1$ and $r_2$ are reverse relations, such as `HusbandOf` and `WifeOf` in the example. Take the case in Figure 2(a) top as an example, the reverse edit makes two updates to the model, but they simultaneously change the fact (Jacques, HusbandOf, ?). Hence, the change can be represented as the following:

$$\begin{cases} k_o = (s_1, r_1, o_2) \\ k_n = (s_2, r_1, o_2) \end{cases} \quad (3)$$

where we can instantiate these two facts as $k_o$:*Marie's husband is Jacques* edited by $e_1$ and $k_n$:*Maurice's husband is Jacques* edited by $e_2$ in the example above. We focus on this kind of fact pair which may lead to confusion when we ask related question *'Jacques is the husband of who?'* to the post-edit model.

**Composite Edit** Furthermore, we explore a more complex situation, where the edits are associated with a fact that will not be influenced by editing. The example illustrated in Figure 2(a) bottom exhibits this situation that we edit $e_1$:*Hamlet was written in English → French* and then $e_2$:*Shakespeare wrote in French → German* while preserving a tied fact $k_f$:*The notable work of Shakespeare is Hamlet*, and we also formalize it as

$$\text{Composite Edit}: \begin{cases} k_f = (s_1, r, s_2) \\ e_1 = (s_1, r_1, o_1 \to o_2) \\ e_2 = (s_2, r_2, o_2 \to o_3) \end{cases} \quad (4)$$

where $r \wedge r_1 \to r_2$ is a logical rule, for example `NotableWork`$\wedge$`WrittenIn`$\to$`Language`. $e_1$ and $e_2$ will both edit the fact (`Hamlet,WrittenIn,?`). Thus, after executing the edits, the confusing knowledge update can be represented as

$$\begin{cases} k_o = (s_1, r_1, o_2) \\ k_n = (s_1, r_1, o_3) \end{cases} \quad (5)$$

After knowledge editing, we expect the post-edit model to answer *German* to the question *What language was Halmet written in?*

### 2.2.2 EVALUATION

To analyze the Knowledge Conflict issue, we design a new dataset CONFLICTEDIT to evaluate the performance of existing knowledge editing methods.

| Method | Single | Coverage | | CONFLICTEDIT | | | | | | |
| | | | | Reverse | | | Composite | | | |
| | Succ↑ | CS↑ | CM↑ | $CS_{exp}$↑ | $CS_{imp}$↑ | CM↑ | $CS_{exp}$↑ | $CS_{imp}$↑ | CM↑ | TFD↓ |
| *GPT2-XL* | | | | | | | | | | |
| FT | 82.56 | 78.88 | 70.86 | 80.28 | 15.20 | **71.11** | 75.45 | 57.65 | **64.28** | 88.75 |
| MEND | 98.40 | 91.04 | 60.01 | **88.89** | **15.32** | 60.50 | **84.85** | **81.35** | 43.45 | 72.09 |
| ROME | 99.96 | **99.76** | **96.92** | 65.92 | 0.00 | -0.65 | 71.70 | 38.70 | 37.04 | 69.55 |
| MEMIT | 79.24 | 83.88 | 32.29 | 51.44 | 2.08 | -1.60 | 57.15 | 29.40 | -1.50 | 24.63 |
| *GPT-J* | | | | | | | | | | |
| FT | 100.0 | **100.0** | **99.90** | **99.60** | 4.16 | **97.20** | 96.68 | 88.92 | 88.98 | 89.97 |
| MEND | 100.0 | 95.88 | 82.41 | 88.92 | **6.40** | 60.72 | 83.04 | 73.52 | 63.99 | 42.95 |
| ROME | 100.0 | 99.80 | 94.25 | 56.84 | 0.00 | 0.06 | 77.60 | 29.24 | 39.27 | 81.02 |
| MEMIT | 100.0 | 99.64 | 88.91 | 55.16 | 0.00 | -1.18 | 75.48 | 49.28 | 28.78 | 64.51 |

Table 1: Knowledge Conflict results of knowledge editing methods. **Bold** results denote the best performance in each situation, whereas red results indicate a total failure under the setup and blue results mark the damage on tied fact that cannot be ignored.

**Setup** We construct our CONFLICTEDIT dataset from WikiData (Vrandecic & Krötzsch, 2014), which consist of two types of edits, namely REVERSE EDIT and COMPOSITE EDIT, as defined in Equation 2 and Equation 4. To begin with, we follow Ho et al. (2020) to obtain several reverse and composite logical rules in WikiData. We construct CONFLICTEDIT by sampling thousands of data of REVERSE EDIT and COMPOSITE EDIT edits respectively. In the COMPOSITE EDIT setup, we respectively utilize GPT2-XL (1.5B) (Radford et al., 2019) and GPT-J (6B) (Wang & Komatsuzaki, 2021) to confirm the correctness of the tied fact $k_f$ before experiments. Also, we use GPT-4 (OpenAI, 2023) to generate neighbor prompts for every relation (Dataset construction details are in Appendix A.1.1). Besides, we take SINGLE and COVERAGE EDIT as references. SINGLE EDIT refers to editing the latest knowledge update directly, which evaluates the model's ability to directly modify the knowledge $(s, r, o_1 \rightarrow o_3)$. COVERAGE EDIT is based on a pair of direct edits sharing the common $(s, r)$, that is

$$\text{Coverage Edit} : \begin{cases} e_1 = (s, r, o_1 \rightarrow o_2) \\ e_2 = (s, r, o_2 \rightarrow o_3) \end{cases} \tag{6}$$

which focuses on a covered knowledge update, referring to the editing target of the last edit $e_2$. We evaluate the result under SINGLE EDIT $e_2$ and COVERAGE EDIT pair $(e_1, e_2)$ as references.

**Metrics** We design a new metric **Conflict Score (CS)**, which weighs how well a knowledge editing method handles the knowledge conflict issue. We measure **CS** by calculating the ratio that the new fact $k_n$ is more possible than the old fact $k_o$ after knowledge editing, that is

$$\text{CS} = \mathbb{1}\{p_{f_{\theta'}}(k_n) > p_{f_{\theta'}}(k_o)\}, \tag{7}$$

$p_{f_{\theta'}}(k)$ is the probability of emitting the target object in $k$ by probing the the prompts of $(s, r)$. Further, we utilize two modes, **Explicit ($CS_{exp}$)** and **Implicit ($CS_{imp}$)**, to calculate $p_{f_{\theta'}}(k_n)$ (Details are in Appendix B). As stated above, the old fact $k_o$ is the reason causing Knowledge Conflict, we design the metric **Conflict Magnitude (CM)** to estimate the decrease of the probability of $k_o$:

$$\text{CM} = \frac{p_{f_{\theta^m}}(k_o) - p_{f_{\theta'}}(k_o)}{p_{f_{\theta^m}}(k_o)}, \tag{8}$$

where $\theta^m$ is the intermediate model parameters after edit $e_1$. For the convenience of analyzing the experimental results and comparing the difference of evaluation paradigm, we take **Success Score (Succ)** as a metric revealing the basic editing performance, that is

$$\text{Succ} = \mathbb{1}\{p_{f_{\theta'}}(o^* \mid (s, r)) > p_{f_{\theta'}}(o \mid (s, r))\}. \tag{9}$$

In the COMPOSITE EDIT setup, the tied fact $k_f$ is also significant to compose the knowledge conflict, thus we consider the probability change of $k_f$ as **Tied Fact Damage (TFD)**, calculated similarly to Equation 8 where we substitute the $k_o$ with $k_f$.

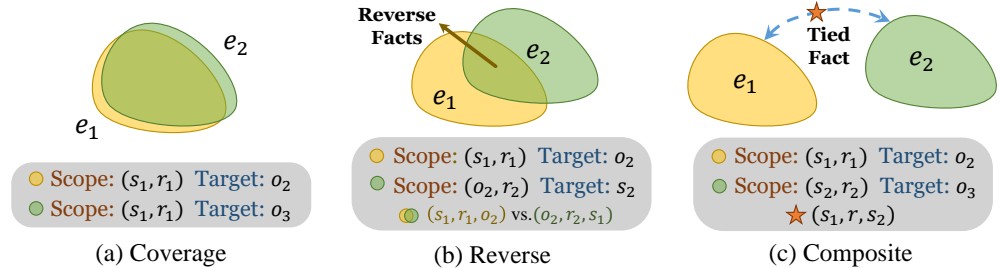

Figure 3: A Unified View of **Knowledge Conflict**. $e_1$ and $e_2$ are two different knowledge editing instances. **Editing Scope** is the range where an edit takes effect, and **Target** is the expected editing object. (a) COVERAGE EDIT shares a total coverage in editing scope. (b) REVERSE EDIT relates each edit through reverse facts. (c) COMPOSITE EDIT is unique in its editing scope yet maintains a consistent logical rule concerning a tied fact denoted as $k_f$.

### 2.2.3 RESULTS ANALYSIS

We conduct experiments on GPT2-XL and GPT-J and report the results in Table 1. In this part, we report the results of SINGLE, COVERAGE EDIT, REVERSE EDIT and COMPOSITE EDIT settings, and further discuss a unified view to expound the internal reason for the results.

**Single & Coverage Edit**  Note that for the COVERAGE EDIT setting, knowledge editing techniques are tasked with not only incorporating new knowledge but also editing existing information to ensure knowledge consistency. The CS scores derived from COVERAGE EDIT can reveal the Reliability and Generalization performance subsequent to implementing a pair of edits. Notably, the results might be influenced by the initial edit within the pair, setting it apart from the Single Succ scores. Empirically, we notice that FT achieves higher scores in GPT-J but lower scores in GPT2-XL because of its disadvantage in generalization referring to the previous observation in Mitchell et al. (2022a). ROME is effective in both GPT2-XL and GPT-J, showing its comparable ability to edit knowledge. MEND and MEMIT obtain lower scores, especially in the CM metric, which indicates the failure of knowledge editing in this case.

**Reverse Edit**  FT and MEND utilize $\mathrm{Prompt}(s_1, r_1, o_1 \to o_2)$ as input, and update model parameters by gradient descent, which is more effective in detecting the update of reverse edit. Thus FT and MEND obtain higher scores on CM which indicates the vanish of the old knowledge $k_o$ through the latest editing. ROME and MEMIT take the subject as a key to attach the edit to all the involved prompts $\mathrm{Prompt}(s_1, r_1)$, thus applying the reverse edit in totally distinct ways. The CS scores indicate a total failure of ROME and MEMIT in this setup, which may be caused by the poor performance on reverse relation reasoning discovered by Berglund et al. (2023). When evaluating the CM metric, both MEND and MEMIT retain the original knowledge within the model, posing a significant challenge to maintaining knowledge consistency in LLMs.

**Composite Edit**  We observe that both FT and MEND demonstrate superior performance in this setup. In contrast, ROME and MEMIT show promising results specifically in the editing process, but the edited facts remain disconnected from other information. Further, we notice that most of previous knowledge editing methods will trigger damage on tied facts with very high TFD score, Note that for the COMPOSITE EDIT setting, knowledge editing approaches are anticipated to modify facts while also taking into account the associations of knowledge already learned within the LLMs.

**A Unified View**  We analyze three types of knowledge editing scopes based on Mitchell et al. (2022b)'s Editing Scope concept to reveal their differences in input space editing. In Figure 3 (a), the COVERAGE EDIT setup presents a binary input space choice with completely overlapping editing scopes. Figure 3 (b) shows that the REVERSE EDIT edits are linked through reverse facts within their scope but completely overlap in logical implication. In Figure 3 (c), COMPOSITE EDIT involves two edits with non-overlapping scopes but establishes a logical chain through an existing tied fact, ensuring logical consistency. This motivates **the employment of detection technologies** for the target knowledge, grounded in the symbolic logical rules of KGs, to circumvent potential knowledge discrepancies.

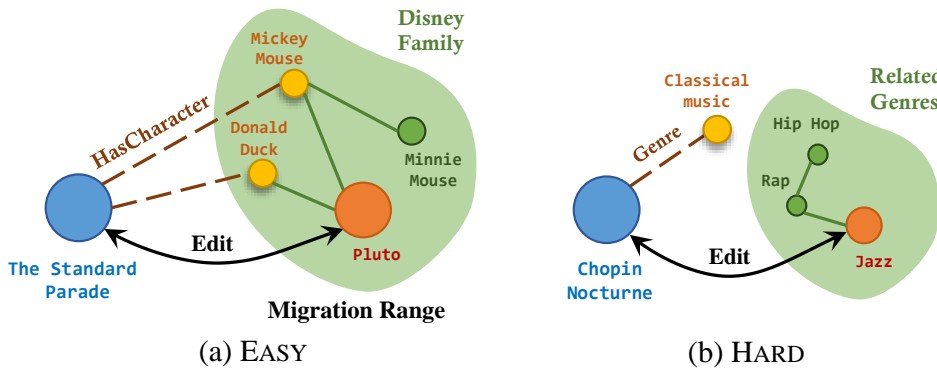

(a) EASY                                    (b) HARD

Figure 4: The EASY and HARD split of ROUNDEDIT. (a) The EASY split contains editing targets which are semantically associated with the true labels of $(s, r)$. The related field is called **Migration Range**. (b) The HARD split edits the object that is irrelevant to the true labels.

## 2.3 KNOWLEDGE DISTORTION ANALYSIS

Despite the capability of current knowledge editing methods to update single factual knowledge, **Knowledge Conflict** reflects the drawbacks of current knowledge editing approaches facing multiple edits. We further analyze whether current knowledge editing approaches will damage the implicit knowledge structure within LLMs or not. Here, we specify the concept of **Knowledge Distortion** and explore the mechanism of how knowledge editing works and impacts LLMs.

### 2.3.1 PROBLEM DEFINITION

**Knowledge Distortion**    As stated in Meng et al. (2022), knowledge editing for LLMs can change the distribution of the next token, thus editing the model's output. Intuitively, fine-tuning on un-balanced data causes preference of certain generations, thus applying one edit at a time seems an extremely unbalanced update to the model. Particularly, in the multi-label scene, for a given relation $r$, there is a set of objects $\text{Obj} = \{o_i; i = 1, 2, 3, 4, ...\}$ that correspond to the subject and relation pair $(s, r)$. When we perform an edit $(s, r, o^* \rightarrow o_1)$ (where $o^* \notin \text{Obj}$), the distribution of next-token probabilities over $\text{Obj}$ tends to favor $o_1$. However, weakening the preference for generating other correct objects in $(s, r, o_i); i = 2, 3, 4, \ldots$ is undesirable for a robust editing method. Unfortunately, most editing methods exhibit a negative impact on the correct objects beyond the target, that is, the knowledge structure for the $(s, r)$ will be distorted.

**Round-Edit**    Note that it is not easy to analyze **Knowledge Distortion** through a single edit, since the distribution of the original model over labels in $\text{Obj}$ is absent to be referred. Therefore, we design a ROUND-EDIT setting which is

$$\text{Round-Edit} : \begin{cases} e_1 = (s, r, o_1 \rightarrow o^*) \\ e_2 = (s, r, o^* \rightarrow o_1) \end{cases} \tag{10}$$

where $o_1$ is the target object, and $o^*$ is the intermediate object. After applying the ROUND-EDIT, we expect that the post-edit model be consistent with the original model or even perform better.

### 2.3.2 EVALUATION

**Setup**    To investigate the performance and mechanism of knowledge distortion caused by knowl-edge editing, we construct a dataset ROUNDEDIT from WikiData, which contains EASY and HARD splits. First of all, we collect several relations that possibly match multiple objects and then filter the subjects. Then we select one object in the object list of a multi-labeled $(s, r)$ as the target ob-ject of ROUND-EDIT. Considering that the objects semantically related to editing target are more likely to be generated by the post-edit model, we introduce two settings EASY and HARD (Dataset construction details are in Appendix A.1.2). As illustrated in Figure 4 (a), the edits in EASY tie an object to a certain subject by migrating a range of similar semantic objects, while HARD edits object that semantically distant from the true labels of $(s, r)$ as depicted in Figure 4 (b). If other labels fall within the Migration Range, the post-edit model not only shows a predilection for the target object but also reinforces its confidence in these other labels. If not, they are overlooked.

| Method | EASY | | | | HARD | | | |
|---|---|---|---|---|---|---|---|---|
| | Succ↑ | D↓ | IR↓ | FR↓ | Succ↑ | D↓ | IR↓ | FR↓ |
| *GPT2-XL* | | | | | | | | |
| FT | 89.50 | 6.47 | 74.47 | 72.24 | 90.06 | 11.38 | 80.83 | 80.82 |
| MEND | 78.22 | 6.48 | 87.86 | 86.88 | 80.50 | 9.73 | 90.56 | 89.36 |
| ROME | 99.82 | 7.78 | 67.41 | 64.60 | 99.86 | 14.86 | 74.38 | 73.68 |
| MEMIT | 86.44 | 5.94 | 49.98 | 45.36 | 88.12 | 10.29 | 53.38 | 50.12 |
| MEMIT+MLE | 83.62 | **3.05** | **4.66** | **1.72** | 86.64 | **2.67** | **2.67** | **1.12** |
| *GPT-J* | | | | | | | | |
| FT | 99.96 | 9.59 | 96.43 | 96.56 | 100.0 | 16.12 | 97.48 | 97.32 |
| MEND | 99.44 | 8.55 | 90.96 | 90.68 | 99.12 | 14.35 | 87.64 | 86.56 |
| ROME | 99.66 | 6.91 | 67.35 | 65.56 | 99.80 | 13.95 | 78.98 | 77.60 |
| MEMIT | 99.52 | 6.44 | 56.91 | 53.52 | 99.72 | 13.50 | 72.03 | 70.44 |
| MEMIT+MLE | 93.96 | **2.11** | **2.48** | **0.80** | 80.34 | **2.72** | **3.84** | **1.12** |

Table 2: Knowledge Distortion results of knowledge editing methods. **Bold** results denote the best performance in each situation.

**Metrics** Inspired by the teacher forcing generation in Meng et al. (2022), we define three new metrics to measure the **Knowledge Distortion** of the edited model. **Distortion (D)** estimates the JS divergence of the distribution on objects in $\text{Obj}$ before and after ROUND-EDITING, that is:

$$D = \text{JS}(p_{f_\theta}(\text{Obj} \mid (s,r)), p_{f_{\theta'}}(\text{Obj} \mid (s,r))), \tag{11}$$

where $D$ denotes the Distortion, $p(\text{Obj} \mid \cdot)$ is a normalized probability distribution over $\text{Obj}$. This metric measures the extent of knowledge distortion. Moreover, inspired by Cohen et al. (2023) we introduce a more specific metric called **Ignore Rate (IR)**. The IR metric quantifies the extent to which the objects in $\text{Obj}$ (excluding the target object $o_1$) are disregarded or overlooked following the process of knowledge editing. IR is calculated as follows:

$$\text{IR} = \frac{1}{|\text{Obj}| - 1} \sum_{\substack{o \in \text{Obj} \\ o \neq o_1}} \mathbb{1}\{p_{f_\theta}(o \mid (s,r)) > p_{f_{\theta'}}(o \mid (s,r))\}, \tag{12}$$

where $p(o \mid (s,r))$ is the generative probability of $o$. Furthermore, we design **Failure Rate (FR)** metric to count the ratio of the case where their $\text{IR} > 0.5$, that is

$$\text{FR} = \mathbb{1}\{\text{IR} > 0.5\}. \tag{13}$$

To make a reference to the basic performance on the dataset, we also calculate the **Success Score (Succ)** as same as §2.2, which computes the average Succ of the edits in ROUND-EDIT.

### 2.3.3 RESULTS ANALYSIS

We conduct experiments on GPT2-XL and GPT-J and then summarize the results in Table 2. The variance of the distribution on $\text{Obj}$ is illustrated in Figure 5 over EASY and HARD settings. Based on the results, we design an effective method Multi-Label Edit (MLE) to address this problem.

**Analysis** As depicted in Table 2, a notable knowledge distortion is observed in FT and MEND, as evidenced by their high IR and FR values. ROME and MEMIT, while still exhibiting some level of distortion, demonstrate lower values compared to FT and MEND. This indicates that ROME and MEMIT have integrated unique mechanisms to minimize disturbances to the implicit knowledge structure in LLMs during knowledge editing. Further, we observe an interesting phenomenon from the EASY and HARD scenarios: when the Succ metric reaches high values, FT and MEND not only underperform but also demonstrate minimal variation across the two datasets; In contrast, ROME and MEMIT not only demonstrate an advantage in absolute values but also show a big gap in IR and FR between EASY and HARD settings. This leads to the conclusion that the editing approaches of ROME and MEMIT effectively achieve scope transfer (better generalization ability) through semantic relations. Moreover, as shown in Figure 5, we illustrate an example of $|\text{Obj}| = 5$ in EASY and HARD respectively to display the inner change of the distribution over $\text{Obj}$. We observe a clear knowledge distortion emerges which inadvertently impacts the LLMs' implicit knowledge structure, particularly when the structure encapsulates a diverse semantic spectrum (e.g., HARD setting).

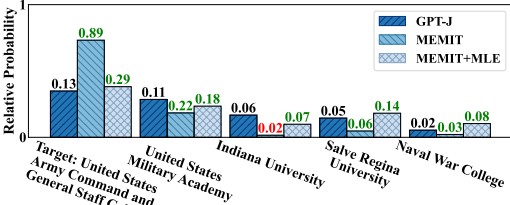 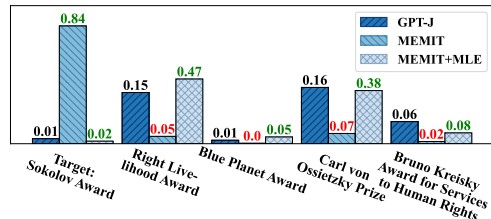

**Input:** James J. Lovelace was educated at ＿.
(a) EASY

**Input:** Uri Avnery received award(s), such as ＿.
(b) HARD

Figure 5: Typical results on sampled cases in EASY and HARD. The height of each bar refers to the generative probability over the 5 true labels of $(s, r)$. Red scores indicate ignorance of true labels, whereas green scores display the positive effects.

**A Simple Solution: Multi-Label Edit (MLE)** To be noted, we aim to evaluate how knowledge distortion affects the generation performance of post-edited models via its impact on IR and FR metrics. The results in Figure 2 indicate that even ROME and MEMIT exhibit poor performance in the EASY scenario, affecting over 50% of the true labels and cases. To this end, we introduce a simple solution: **Multi-Label Edit (MLE)**, tailored for such multi-label scenarios. Specifically, as shown in Figure 2, we take three objects (one-to-many triples taken from existing KG which is included in our datasets) as the editing target, expecting to preserve the performance over the true labels (More details in Appendix C). From the results in Figure 2, it is evident that MLE can help mitigate the influence of knowledge distortion, which is also demonstrated by cases in Figure 5.

## 3 RELATED WORK

Recently, there has been a surge in knowledge analysis and editing for LLMs. This encompasses efforts to demystify the knowledge storage mechanisms inherent in these "black-box" neural networks (Geva et al., 2021; Dai et al., 2022). Researchers have delved into deconstructing the Transformer architecture by exploring the concept of knowledge/skill neurons (Dai et al., 2022; Chen et al., 2023b; Wang et al., 2022), and complex system science (Holtzman et al., 2023). Additionally, there's a keen interest in developing effective strategies to edit knowledge within LLMs (Xu et al., 2023; Hase et al., 2023; Hartvigsen et al., 2022; Cheng et al., 2023; Hernandez et al., 2023; Li et al., 2023a; Wang et al., 2023a; Xu et al., 2023; Yu et al., 2023; Wang et al., 2023c; Li et al., 2024; Huang et al., 2024; Ma et al., 2024a; Song et al., 2024; Ma et al., 2024b; Peng et al., 2024; Wei et al., 2024; Gu et al., 2024). More efforts are devoted to investigating the completeness and robustness of these methods. In order to determine the exact scope of effectiveness when applying certain edit, it is crucial to explore evaluation samples that are correlated with the edits (Yin et al., 2023; Cohen et al., 2023; Wu et al., 2023; Hoelscher-Obermaier et al., 2023; Wei et al., 2023; Onoe et al., 2023; Wang et al., 2023c; Ma et al., 2023). Simultaneously, knowledge editing injects new facts into LLMs, thus verifying the connections or reasoning among the new knowledge and the pre-existing knowledge in the model even multiple new knowledge has become a prevalent topic (Zhong et al., 2023; Li et al., 2023b; Hua et al., 2024; Ju et al., 2024; Hua et al., 2024). Furthermore, recent work has also begun to address the catastrophic forgetting issue (Gupta et al., 2024) caused by knowledge editing and the problem of guiding language models to generate harmful content (Hazra et al., 2024).

## 4 DISCUSSION AND CONCLUSION

This paper illustrates that knowledge editing can bring in side effects of knowledge conflict, leading to knowledge inconsistency and may even enhance the hallucination of LLMs (Ji et al., 2023; Rawte et al., 2023; Zhang et al., 2023). We argue that conflict detection via logical rules or KG reasoning may avoid knowledge conflict. However, the side effect of knowledge editing for LLMs is far from satisfactory, which needs efforts for future works. Furthermore, knowledge editing technologies can serve as a scaffold to update, sanitize, or personalize LLMs (Ishibashi & Shimodaira, 2023; Chen et al., 2023a; Mao et al., 2023). Empirical observation of knowledge distortion in this paper brings about a severe issue to updating knowledge in LLMs. Although we introduce Multi-Label Edit to mitigate knowledge distortion, more work should be developed to avoid the problem.

## ACKNOWLEDGMENTS

We would like to express gratitude to the anonymous reviewers for kind comments and MEMIT (Meng et al., 2023), EasyEdit (Wang et al., 2023b) and many other related works for their open-source contributions as well as GPT-4 Service (OpenAI, 2023). This work was supported by the National Natural Science Foundation of China (No. 62206246), the Fundamental Research Funds for the Central Universities (226-2023-00138), Zhejiang Provincial Natural Science Foundation of China (No. LGG22F030011), Ningbo Natural Science Foundation (2021J190), CAAI-Huawei MindSpore Open Fund, Yongjiang Talent Introduction Programme (2021A-156-G), CCF-Tencent Rhino-Bird Open Research Fund, and Information Technology Center and State Key Lab of CAD&CG, Zhejiang University.

## ETHICAL CONSIDERATIONS

The datasets and models presented in this paper are designed for exploratory analysis of LLMs. It is crucial to note that data from WikiData isn't always error-free, and GPT-4 might also introduce biases. As a result, when constructing datasets of knowledge editing for LLMs, there is a risk of surfacing knowledge infused with offensive language or prejudicial content. During our experiments, **we have carefully reviewed all data, ensuring the removal of any toxic or offensive content**.

## REPRODUCIBILITY STATEMENT

Codes and datasets are available at https://github.com/zjunlp/PitfallsKnowledgeEditing. We provide technical details of dataset construction including CONFLICTEDITand ROUNDEDITin Appendix A.1.1 and Appendix A.1.2, respectively. We provide detailed experimental settings in Appendix B. We provide more technical details of Multi-Label Edit (MLE) in Appendix C.

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

# A DATASETS DETAILS

## A.1 CONSTRUCTION OF DATASETS

### A.1.1 CONFLICTEDIT

We construct our CONFLICTEDIT dataset based on WikiData (Vrandecic & Krötzsch, 2014), using the raw data and some logical rules mined from it. From the original WikiData data to the CONFLICTEDIT dataset, we have taken the following steps:

**Step 1** First, we determine the binary and ternary logic rules we need using the method of Ho et al. (2020) and manually select rules/relations, thereby obtaining the candidate sets of relations in REVERSE EDIT and COMPOSITE EDIT, respectively.

**Step 2** Next, based on the nominated relations and corresponding logical rules, we construct Wiki-Data Queries and use the WikiData Query Service to mine an average of 50,000 factual combinations that meet the requirements for each logical rule.

**Step 3** We sample from each factual combination present in the logical rules, excluding those with descriptions in numeric formats or those with identical entities. To ensure accuracy, we validate the integrity of the associated fact while constructing the COMPOSITE EDIT dataset for each model (e.g., GPT-XL, GPT-J). Specifically, we use the prompt associated with the fact as an input, prompting the model to generate outputs of a predetermined length using the Top-1 next-token generation approach. We then verify whether these outputs include the object of the fact.

**Step 4** Finally, we select 2,500 data points each from binary and ternary factual combinations. However, due to GPT-2 XL's comparatively weaker performance in verifying the accuracy of tied facts against GPT-J, we only select 2,000 data points for it. We then sample edited target entities under each relation to assemble the final REVERSE EDIT and COMPOSITE EDIT datasets. Additionally, using samples from both datasets, we create the COVERAGE EDIT dataset to serve as an experimental benchmark.

**Step 5** We've manually crafted editing prompts for each relation, drawing inspiration from their descriptions on WikiData. Based on this, we employ GPT-4 (OpenAI, 2023) to generate 20 alternative phrasings for each relation's prompt. From these, we handpick the 10 most appropriate prompts for our generalization evaluation.

For training MEND, samples with locality facts are essential. Prioritizing training efficacy, we opt not to utilize the NQ dataset, as done by ZsRE. Instead, our sampling is anchored in COUNTERFACT (Meng et al., 2022). Table 3 illustrates the data structure we've established for both REVERSE EDIT and COMPOSITE EDIT, also highlighting a specific example from COMPOSITE EDIT.

| | |
|---|---|
| $\mathcal{R}$ | Mother∧Spouse→Father |
| $\mathcal{F}$ | (Philip Leakey, Mother, Mary Leakey)
(Mary Leakey, Spouse, Louis Leakey)
(Philip Leakey, Father, Louis Leakey) |
| $\mathcal{E}$ | $e_1$: (Mary Leakey, Spouse, Louis Leakey → Mary Campbell of Mamore)
$e_2$: (Philip Leakey, Father, Mary Campbell of Mamore → Andres Ehin) |
| $k_f$ | (Philip Leakey, Mother, Mary Leakey) |
| $k_o$
$k_n$ | (Mary Leakey, Spouse, Mary Campbell of Mamore)
(Mary Leakey, Spouse, Andres Ehin) |

Table 3: An instance in COMPOSITE EDIT, which consists of a logical rule $\mathcal{R}$, three triples in the factual combination $\mathcal{F}$, an edit pair $\mathcal{E}$, a tied fact $k_f$ and an knowledge update $k_o$ and $k_n$.

| Dataset Split | Relation Type | Input | Intermediate Object | True Labels |
|---|---|---|---|---|
| EASY | 1-5 `EducatedAt` | James J. Lovelace was educated at … | Cranbrook Educational Community | **Target: United States Army Command and General Staff College**, United States Military Academy, Indiana University, Salve Regina University, Naval War College |
| HARD | 1-5 `AwardReceived` | Uri Avnery received award(s), such as … | Hero of the Mongolian People's Republic | **Target: Sokolov Award**, Right Livelihood Award, Blue Planet Award, Carl von Ossietzky Prize, Bruno Kreisky Award for Services to Human Rights |

Table 4: Examples in EASY and HARD corresponding to the results in Figure 5.

### A.1.2 ROUNDEDIT

We construct ROUNDEDIT dataset using data sourced from WikiData, enriched by some 1-to-n relations mined from it. Transitioning from the raw WikiData data to the ROUNDEDIT dataset required several stages:

**Step 1**  We first manually select some 1-to-n relations and construct WikiData Queries, utilizing the WikiData Query Service to mine 5000 $(s, r)$ pairs along with the correct object set $\text{Obj}$. From each relation, we selectively sample, ensuring that the 'n' value in the 1-n relations remains below 10. Furthermore, we exclude samples with descriptions in a numeric format.

**Step 2**  For each LLM, we perform next-token generation tests on the raw data samples, filtering these samples based on generation probabilities. To delineate the EASY and HARD dataset categories, we examine the correlation between semantic relevance and generation probabilities. Using this analysis, we measure the semantic relationship among the correct labels and subsequently sample to establish both the EASY and HARD categories, each comprising 2,500 data points.

**Step 3**  Similarly, we manually create editing prompts for every relation, drawing from their respective descriptions on WikiData. Based on this, we utilize GPT-4 (OpenAI, 2023) to generate 20 rephrased prompts. Of these, we handpick the top 10 prompts, incorporating them into the generalization evaluation.

We also sample locality facts from COUNTERFACT (Meng et al., 2022) to train the MEND. Figure 4 showcases selected examples from the EASY and HARD settings.

## B  IMPLEMENTING DETAILS

**FT**  For basic Fine-Tuning (FT), we follow Meng et al. (2022) to re-implement their study, which uses Adam (Kingma & Ba, 2014) with early stopping to minimize $-log\mathbb{P}_{G'}[o* \mid p]$, changing only $mlp_{proj}$ weights at selected layer 1 in GPT2-XL and layer 21 in GPT-J. For both models, all hyper-parameters follow default settings. To ensure fairness in the experiments, we always use the unconstrained fine-tuning approach.

**MEND**  Mitchell et al. (2022a) develops an efficient method for locally editing language models using just a single input-output pair. Essentially, MEND employs a technique to manipulate the gradient of fine-tuned language models which leverages a low-rank decomposition of gradients. Here we adopt the COUNTERFACT (Meng et al., 2022) to evaluate the locality while training MEND. For convenience, we train MEND respectively on our CONFLICTEDIT and ROUNDEDIT datasets using EasyEdit (Wang et al., 2023b). The hyper-parameters follow default settings both on GPT2-XL and GPT-J.

**ROME**  ROME, as proposed by Meng et al. (2022), conceptualizes the MLP module as a straightforward key-value store. For instance, if the key represents a subject and the value encapsulates

knowledge about that subject, then the MLP can reestablish the association by retrieving the value that corresponds to the key. In order to add a new key-value pair, ROME applies a rank-one modification to the weights of the MLP, effectively injecting the new information directly. This method allows for more precise and direct modification of the model's knowledge through knowledge locating. We directly apply the code and MLP weight provided by the original paper and keep the default setting for hyper-parameters.

**MEMIT**  MEMIT (Meng et al., 2023) builds upon ROME to insert many memories by modifying the MLP weights of a range of critical layers. We test the ability of MEMIT using their source code and all hyper-parameters follow the same default settings.

**Explicit (CSexp) and Implicit (CSimp)**  Here, 'Explicit' and 'Implicit' are in reference to the second edit. When computing $p_{f_{\theta'}}(k_n)$, we can utilize the edit target $(s_2, r_2, o_2^*)$ from the second edit $e_2 : (s_2, r_2, o_2 \rightarrow o_2^*)$ as the input for calculating $k_n$, which we term the Explicit mode. Conversely, we employ a more relevant latent target associated with the first edit as the input for computing $k_n$, and this mode is referred to as the Implicit mode. Table 5 lists the edit tatget of each dataset split in every mode.

| Dataset Split | Explicit Mode | Implicit Mode |
|---|---|---|
| Coverage | $(s_2, r_2, o_2^*)$ | $(s_1, r_1, o_2^*)$ |
| Reverse | $(s_2, r_2, o_2^*)$ | $(o_2^*, r_1, s_2)$ |
| Composite | $(s_2, r_2, o_2^*)$ | $(s_1, r_1, o_2^*)$ |

Table 5: Explicit (CSexp) and Implicit (CSimp) of each dataset split.

## C  DETAILS OF MULTI-LABEL EDIT

To address the issue of Knowledge Distortion in knowledge editing, we introduce a novel approach termed **Multi-Label Editing (MLE)**. This method diverges from conventional knowledge editing. Our objective is to simultaneously edit groups of knowledge that possess the same $(s, r)$ into the model, leveraging MEMIT's multi-editing capabilities during the process. For instance, when we make an edit *Joe Biden was born in California → Florida*, we can consider the differences in the knowledge structures of California and Florida, recall their related semantics within the model (such as America), and then regard 'Florida' and 'America' as labels utilizing MLE to edit.

Our findings indicate that such a concurrent editing approach not only prevents unnecessary knowledge updates but also ensures the model doesn't favor a particular label. Through rigorous experimental results, we demonstrate that MLE serves as an effective strategy to mitigate knowledge distortion issues. By identifying and incorporating related or essential knowledge within the model prior to the editing of a specific piece of knowledge, we can reduce the potential disruption to the LLM's inherent knowledge structure caused by the editing process.

