# OpenReview forum: "Unveiling the Pitfalls of Knowledge Editing for Large Language Models"
_ICLR.cc/2024/Conference — ICLR 2024 poster_

### Official Review · Reviewer_YAyi · 2023-10-30

**Soundness:** 2 fair
**Presentation:** 3 good
**Contribution:** 2 fair
**Rating:** 6
**Confidence:** 3

**Summary:**

This paper studies the potential pitfalls related to LLMs knowledge editing, including Knowledge Conflict and Knowledge Distortion.
To achieve this target, two benchmark datasets and several innovative evaluation metrics are also introduced.
With these settings, this paper conducts experiments from two aspects among four common editing approaches on two LLMs, and proposes a Multi-Label Edit (MLE) solution.

**Strengths:**

1. This paper puts forward a valuable research problem, i.e., to discuss the potential risk of editing knowledge encoded in LLM. I believe this could help researchers and practitioners better understand and manipulate the knowledge encoded in LLM, so as to obtain a better model.
2. Extensive baselines are employed for analyzing the proposed research question.
3. Several new measures are designed to quantify the degree of knowledge conflicts in experiments.

**Weaknesses:**

My major concerns lie in the following three aspects:
1. The “Knowledge Conflict” proposed in this paper is confusing to me.
2. The experimental settings of “Knowledge Distortion” are vague and incomplete.
3. The proposed MLE solution is unclear.

**Questions:**

My questions mainly comes from the above three concerns.
1. The knowledge conflict mentioned in your paper, i.e., “e1: Marie’s husband is Pierre → Jacques and e2: Jacques’s wife is Marie → Maurice”, seems to be the “Data Collision” rather than “Editing Conflict”.
If this type of conflict only exists at the data level, then the evaluation in this paper is meaningless.
In addition, I guess you want to emphasize the conflict between different editing operations, as mentioned in Section 2.2 “there is a possibility that interference occurs between different edits, causing the former edit invalid.”.
Could you further clarify this problem and provide more suitable examples?

2. The “Knowledge Distortion” is a very promising research question that I pay attention to. However, the evaluation in this aspect is:
a) vague:
How many the (s,r) pairs did you evaluate to calculate the results in Table 2 ? (Is there five?)
How are the values in Table 2 calculated? Are they arithmetic mean values?
Why is the JS divergence chosen? Is the asymmetric KL divergence inappropriate? and why?

b) incomplete:
Why only evaluate triples under the same (s, r), and other knowledge is not affected? (e.g, with same (s,.,.) and same (., r, .) or others (., ., .))?

3. The proposed MLE method is unclear. Could you explain what the multi-label is and what is its function?
How does this method alleviate the problem of “Knowledge Distortion”?
Additionally, is this method effective for the first question ("Knowledge Conflict") mentioned in your paper?

---

> ### Author Response · Authors · 2023-11-19
> **Response to Reviewer YAyi (Part 1/2)**
>
> We appreciate the time you dedicated to reviewing our work. Below, we would like to provide further clarification.
>
> ---
>
> > **The Essence of Knowledge Conflict**
>
> As the number of edit samples increases, we're concerned about the potential correlation between subsequent edits and those previously applied. If the editing process overlooks the correlation between edit samples and if these samples point towards different or even opposing editing targets, it could introduce conflicts into the model, resulting in a Knowledge Conflict issue. **In essence, Knowledge Conflict arises from logical overlaps in multiple edits during the editing process, leading to incomplete internal knowledge updates within the model and resulting in inconsistency.** We will further explain this definition in the following response.
>
> ---
>
> > **“e1: Marie’s husband is Pierre → Jacques and e2: Jacques’s wife is Marie → Maurice”, seems to be the “Data Collision” rather than “Editing Conflict”…Could you further clarify this problem and provide more suitable examples?**
>
> Intuitively, the Coverage Edit appears as a complete Editing Conflict. However, this type of Editing Conflict doesn't necessarily cause what we refer to as Knowledge Conflict. If, after applying $e_1: (s, r, o_1 \rightarrow o_2)$, the model can update the results of (s, r, ?) to $o_2$, and then, upon applying $e_2: (s, r, o_2 \rightarrow o_3)$, further update the outcome edited by $e_1$ to $o_3$, then there wouldn't be an inconsistency in the model's knowledge on this matter. If, however, after applying $e_2$, the model retains the results from $e_1$, it creates uncertainty in the predictions for s and r, which we term as Knowledge Conflict.
>
> **Understanding the scenario of Reverse Edit might be more challenging, but its essence is similar to the aforementioned Coverage Edit.** From Coverage Edit to Reverse Edit, and finally to the Composite Edit, we illustrate their differences in Editing Scope in Figure 2. From the perspective of Editing Conflict you mentioned, **Composite Edit entirely lacks any Editing Conflict. However, it may still lead to internal knowledge inconsistency**, marking the fundamental distinction between Editing Conflict and Knowledge Conflict.
>
> ---
>
> > **The “Knowledge Distortion” is a very promising research question that I pay attention to. However, the evaluation in this aspect is: a) vague: How many the (s,r) pairs did you evaluate to calculate the results in Table 2 ? (Is there five?) How are the values in Table 2 calculated? Are they arithmetic mean values? Why is the JS divergence chosen? Is the asymmetric KL divergence inappropriate? and why?**
>
> We apologize for causing confusion in some implementation details. Here, we'll address your queries, and we plan to supplement related content in the main text or appendix.
>
> **Dataset Detail:** We utilized 2500 (s, r) pairs to derive the results in Table 2, covering relations samples from 1-to-2 to 1-to-10.
>
> **Experiment Detail:** The results in Table 2 are based on arithmetic mean values among the samples. Regarding the choice of JS divergence over KL divergence, we simply believe that when considering differences between two distributions, asymmetry is unnecessary. Additionally, computation of JS divergence is more efficient.

---

> ### Author Response · Authors · 2023-11-19
> **Response to Reviewer YAyi (Part 2/2)**
>
> > **Why only evaluate triples under the same (s, r), and other knowledge is not affected? (e.g, with same (s,.,.) and same (., r, .) or others (., ., .))?**
>
> In the "Vanilla Evaluation" mentioned in $\S$2.1, **Locality is generally used to evaluate performance on $(\cdot, r, \cdot)$ and $(\cdot, \cdot, \cdot)$.** The paper didn't further conduct experiments on this metric. Additionally, **there's still some controversy in the evaluation on $(s, \cdot, \cdot)$**. In some instances, we believe that other relations with the same s (e.g., born in and mother tone) are correlated, and the model's performance on these relations should also update as part of demonstrating generalization. However, with certain relations (e.g., born in and married with) having no correlation, the evaluation of these triplets forms a part of Locality assessment. Hence, this paper only considers cases with the same s, r pair.
>
> ---
>
> > **Could you explain what the multi-label is and what is its function? How does this method alleviate the problem of “Knowledge Distortion”? Additionally, is this method effective for the first question ("Knowledge Conflict") mentioned in your paper?**
>
> We also apologize for the confusion in this section and will provide an explanation here. We plan to update this part in the subsequent revision to enhance understanding of our research.
>
> **Multi-Label Edit (MLE):** In our dataset, "Multi-Label" refers to the "n" correct objects in the mentioned 1-to-n relations. Due to the distortion in prediction distributions before and after editing among these objects, the model exhibits a bias toward edited objects while completely ignoring other correct objects. Our MLE method is a theoretical attempt to address this issue. Its conclusion is that by simultaneously editing multiple correct objects, we maintain the structure of the 1-to-n relation post-editing. **We believe that MLE fundamentally mitigates the issue of sample imbalance introduced when editing a single sample.** Unfortunately, this method doesn't address the Knowledge Conflict problem as it discusses knowledge update issues unrelated to the model's internal knowledge structure.
>
> We sincerely hope the above answers could resolve your concerns!

---

> ### Comment · Reviewer_YAyi · 2023-11-22
>
> Thanks the authors for their reply. I have read the authors' response and decide to increase the rating a bit. Some issues are remaining:
>
> (1). I find it challenging to fully grasp the concept of "editing conflict" as mentioned in your response. On the one hand, this concept in your rebuttal is articulated as "the conflicts caused by ignoring relevance between samples" (i.e., see the sentence “In essence, …” in your response). However, it appears to be later translated into the "ill-result caused by an unsuccessful editing operation" (i.e., “…the model retains the results from e1…”). This transition is still confusing to me.
> (2). Even the proposed “conflict” is valid, the significance of the “editing conflict” is still questionable. Specifically, even if there is a conflict between two different editing, this is a natural phenomenon; because there probably be a conflict between “old” and “new” knowledge and it is the point of knowledge editing. From this point of view, the “conflict” mentioned in your submission may be meaningless. Furthermore, as mentioned in your response, the proposed MLE method could not solve the "knowledge conflict" issue. Hence, it seems that the contribution of this submission is limited.

---

> > ### Author Response · Authors · 2023-11-22
> > **Response to Reviewer YAyi**
> >
> > Thanks for your careful review and response. Here, I will supplement the previous reply in the hope of further addressing your issues.
> >
> > ---
> >
> > > **The differences and connections between Editing Conflict and Knowledge Conflict**
> >
> > **Editing Conflict**: This concept has not been introduced in our paper. **For the sake of understanding conflicts at the data level and conflicts within the internal knowledge of models, we use this concept as a transition.** We can provide a more explicit definition: **Editing Conflict refers to overlaps at the data level between edited samples** (which might align with what you mentioned as "Data Collision"). For instance, your example of "e1: Marie’s husband is Pierre → Jacques and e2: Jacques’s wife is Marie → Maurice" illustrates a form of reverse overlap. In this regard, the Coverage Edit stands as an extreme case, where this conflict was already known before the application of editing.
> >
> > **Knowledge Conflict**: This concept no longer addresses conflicts in edit data; **its essence lies in whether editing methods can handle Editing Conflict situations correctly.** For instance, in the case of Coverage Edit, we aim for the editing process of (s, r, ?) to be $o_1\rightarrow o_2 \rightarrow o_3$. However, if the editing method is unaware of the relationship between edit samples and retains both editing targets, $o_2$ and $o_3$ within the model **(even the two edits are successful in their own evaluation)**, it leads to inconsistency in the internal knowledge (e.g., in the case of Reverse Edit, two edits may result in a situation where a's husband is b, but b's wife is c). We term this inconsistency Knowledge Conflict.
> >
> > ---
> >
> > > **The solution for Knowledge Conflict**
> >
> > As you mentioned, MLE indeed addresses only Knowledge Distortion but not Knowledge Conflict. We believe that resolving Knowledge Conflict issues comprehensively requires more robust knowledge editing methods. From a practical application perspective, we can mitigate this problem through some small tricks, such as pre-using a logical rule library for data augmentation on edit samples, ensuring that each edit affects inputs logically related. However, with the expansion of this influence scope, we are concerned that it might potentially lead to other issues.

---

### Official Review · Reviewer_XF6v · 2023-10-31

**Soundness:** 2 fair
**Presentation:** 2 fair
**Contribution:** 2 fair
**Rating:** 6
**Confidence:** 3

**Summary:**

This work pioneers the investigation into the potential pitfalls associated with knowledge editing for LLMs, and introduces new
benchmark datasets to evaluate LLMs after knowledge finetuning with proposed innovative evaluation metrics.

**Strengths:**

1. Novel benchmarks and evaluation metrics are developed in the paper
2. With empirical analysis, the authors develop a simple method, a.k.a Multi-Label Edit, to alleviate Knowledge Distortion in LLMs

**Weaknesses:**

1. The novelty of the developed method is quite low and the real contribution of this paper is the development of new benchmarks equipped with evaluation metrics.

**Questions:**

Thanks for your efforts in investigating two pivotal concerns in LLMs, specifically Knowledge Conflict and Knowledge Distortion, which has been widely discussed in NLP community nowadays. However, there remains some issues that I need to discuss with you.


1. Fig. 2 illustrates that after the process of Round-Edit, LLMs tend to assign higher probabilities to the knowledge facts stored in recent corpus and gradually forget the knowledge stored in model parameters. Nonetheless, I believe that the demonstration of Knowledge Distortion in Fig. 2 is not a unique issue limited to LLMs, but rather a prevalent concern across all current Deep Learning models.
And I believe that the solution to this issue is to develop more reasonable retrieval-based LLMs, which answers questions based on the knowledge context, rather than finetune-based Knowledge Editing methods mentioned in this article. With retrieval-based LLMs, you just need to modify the knowledge facts stored in the context, then LLMs can directly answer the user question according to the context. In such consideration, I suppose the contribution of this paper is limited, and it will be better to include evaluation of retrieval-based LLMs as baselines.

2. The novelty of devlelopment of Multi-Label Edit is quite low, why not consider memory-based methods or EMA methods to alleviate the forgetting of previous knowledge stored in LLMs when you are certain that these knowledge fact are all accurate.

3. For Knowledge Conflict, it will also be a problem in retrieval-based LLMs, any thoughts to solve this problem according to your expiermental results?

---

> ### Author Response · Authors · 2023-11-19
> **Response to Reviewer XF6v (Part 1/2)**
>
> We would like to thank you very much for the detailed feedback and valuable suggestions. we hope the following comments could address your questions.
>
> ---
>
> > **The Essence of Knowledge Distortion**
>
> The results obtained from evaluating the impact of knowledge editing methods on unrelated samples using the locality↑ metric after editing a single piece of knowledge with the current knowledge editing methods are as follows [1]:
>
> | Model (Dataset) | FT | MEND | ROME | MEMIT |
> | --- | --- | --- | --- | --- |
> | **GPT-J (ZsRE)** | 37.24 | 97.39 | 99.19 | 99.62 |
> | **GPT-J (CounterFact)** | 1.02 | 93.75 | 93.61 | 97.17 |
>
> Based on the results from this table, we can see that the current knowledge editing methods differ from fine-tuning, as editing a single knowledge has a relatively minor impact on irrelevant knowledge within the model.
>
> **The Knowledge Distortion issue proposed in this paper, unlike the influence on irrelevant knowledge mentioned above, essentially aims to explore the degradation of the original knowledge structure due to the imbalance in samples under specific conditions.**
>
> In investigating the Knowledge Distortion problem, **we utilized the Round-Edit strategy to differentiate it from the normal generalization process of knowledge editing.** According to the definition of knowledge editing generalization, when applying the edited sample "Joe Biden was born in California → Vienna," the answer to the question "What country was Joe Biden born in?" would change from the USA to Austria. In this example, our focus shouldn't be on maintaining knowledge structure related to California but rather on emphasizing the knowledge structure related to Vienna. Therefore, evaluating the Knowledge Distortion problem becomes challenging. Through Round-Edit, we completely disregard generalization concerns and can assess the impact of knowledge editing on the model's knowledge structure.
>
> ---
>
> > **I believe that the solution to this issue is to develop more reasonable retrieval-based LLMs…In such consideration, I suppose the contribution of this paper is limited, and it will be better to include evaluation of retrieval-based LLMs as baselines.**
>
> As you said, retrieval-based knowledge editing methods do indeed possess stronger interpretability in many settings. **However, existing retrieval-based knowledge editing methods like SERAC [2] and IKE [3] currently lack support for generating Knowledge Conflict and settings conducive to studying Knowledge Distortion in this paper.** Our focus is on the interaction of multiple edits. SERAC, due to limitations in its training process, can only handle single edit retrieval scenarios. On the other hand, the Memory+IKE (Retrieval-based LLMs) approach incorporates multiple edit samples as in-context inputs into the model, but strictly speaking, this cannot be considered as the impact of editing methods on the model's internal structure. We believe that both of these are unable to offer additional reference value while regarded as baselines.

---

> ### Author Response · Authors · 2023-11-19
> **Response to Reviewer XF6v (Part 2/2)**
>
> > **The novelty of devlelopment of Multi-Label Edit is quite low, why not consider memory-based methods or EMA methods to alleviate the forgetting of previous knowledge stored in LLMs when you are certain that these knowledge fact are all accurate.**
>
> 1. The retrieval-based knowledge editing method offers stronger interpretability, while fine-tuning-based editing doesn’t require additional parameters or entail reasoning costs, thus holding significant practical value. However, in our actual usage, we haven't effectively integrated the advantages of retrieval-based and fine-tuning-based editing methods to address the Knowledge Distortion issues mentioned in this paper.
>
> 2. The Multi-Label Editing (MLE) method we propose emperically alleviates knowledge distortion problems. However, we can't precisely define under a specific editing sample what needs generalization transfer and what should be preserved. **Nevertheless, MLE offers an approach to tackle this issue. For instance, in the example "Joe Biden was born in California → Vienna," we can consider the differences in the knowledge structures of California and Vienna, recall their related semantics within the model [4], and select the semantics that need to be retained.**
>
> ---
>
> > **For Knowledge Conflict, it will also be a problem in retrieval-based LLMs, any thoughts to solve this problem according to your experimental results?**
>
> It's important to note that the Knowledge Conflict discussed in retrieval-based Language Model (LLMs) primarily revolves around conflicting facts retrieved by the model. This differs slightly in concept from the knowledge conflicts resulting from inadequate model knowledge updates discussed in this paper. **From the experimental results presented in this paper, we glean some insights. For instance, during the retrieval process, employing logical rules to identify factors that might lead to uncertainty of model predictions or hallucination could be beneficial.**
>
> Hopefully, these insights could address your questions.
>
> ---
>
> *[1] Editing Large Language Models: Problems, Methods, and Opportunities (EMNLP 2023)*
>
> *[2] Memory-Based Model Editing at Scale (ICML 2022)*
>
> *[3] Can We Edit Factual Knowledge by In-Context Learning?*
>
> *[4] Dissecting Recall of Factual Associations in Auto-Regressive Language Models (EMNLP 2023)*

---

> ### Author Response · Authors · 2023-11-23
> **Gentle Reminder and Appreciation for Continued Participation in Manuscript Review Discussion to Reviewer XF6v**
>
> Dear Reviewer XF6v,
>
> We would like to express our profound gratitude for your insightful review. Following your valuable suggestions, we have replied to your issues and updated our submission.
>
> Could we kindly enquire if our responses and adjustments have adequately resolved your concerns? We are more than happy to answer any further queries or concerns you may have. Thank you once again.
>
> Best Regards,
>
> Authors

---

> > ### Comment · Reviewer_XF6v · 2023-12-04
> >
> > Thanks, the response of authors has solved my major concerns and I would like to raise my score.

---

### Official Review · Reviewer_ihfd · 2023-10-31

**Soundness:** 3 good
**Presentation:** 3 good
**Contribution:** 3 good
**Rating:** 8
**Confidence:** 4

**Summary:**

This paper explores the potential pitfalls of knowledge editing for Large Language Models (LLMs). It introduces new benchmark datasets and evaluation metrics to investigate the issues of knowledge conflict and knowledge distortion. The results demonstrate that knowledge editing can lead to unintended consequences and inconsistencies in LLMs. The paper also presents potential solutions and challenges for knowledge editing in LLMs.

**Strengths:**

1. The information provides insights into different knowledge editing methods and their performance in various setups.
2. The paper discusses the concept of knowledge distortion and its impact on language models.
3. The paper introduces the idea of conflict detection technologies to address potential knowledge discrepancies.

**Weaknesses:**

1. The information provided is quite technical and may be difficult for non-experts to understand.
2. Some sentences are poorly structured and difficult to comprehend.

**Questions:**

1. What do you think is the fundamental reason for knowledge distortion during knowledge editing for LLMs? How to handle cases of one-to-many knowledge editing?
2. How do the knowledge editing methods compare to each other in terms of their effectiveness and efficiency, what is the takeaway in method selection?
3. In Multi-label Edit, how to guarantee the overall conceptual hierarchy among labels?

Minor Issues:
1. "Emperically" should be "Empirically."
2. "ROME is effective in both GPT-XL and GPT-J" should be "ROME is effective in both GPT2-XL and GPT-J."
3. "This motivates us to employ conflict detection technologies" should be "This motivates the employment of conflict detection technologies."
4. "Knowledge Conflict has reflected" should be "Knowledge Conflict reflects."
5. "However, it is undesirable for a robust editing method to weaken the preference" should be "However, weakening the preference is undesirable for a robust editing method."

---

> ### Author Response · Authors · 2023-11-19
> **Response to Reviewer ihfd**
>
> We sincerely appreciate the time and effort you dedicated to reviewing our work. Your constructive comments are much appreciated, and we would like to address each of the questions you raised below.
>
> ---
>
> > **What do you think is the fundamental reason for knowledge distortion during knowledge editing for LLMs? How to handle cases of one-to-many knowledge editing?**
>
> 1. The primary reason lies in the reliablity and generalization target of traditional knowledge editing evaluation methods, which only ensure the expected performance of the model on edited and similar samples but **overlook the issue of sample imbalance introduced by editing a single sample**.
> **Our proposed Multi-Label Editing (MLE) offers insights to address this issue in editing scenario of one-to-many relations.** Experimental results of MLE demonstrate empirically that by initially mining knowledge structures related to the edited sample (or some relevant tokens) from the original model, knowledge editing can alleviate distortion.
>
> 2. **However, our efforts in addressing Knowledge Distortion are not yet sufficient to propose an application scheme coherent with generalization metric for knowledge editing.** According to the definition of generalization, after applying the edit sample "Joe Biden was born in California → Vienna," the answer to the question "What country was Joe Biden born in?" would change from the USA to Austria. In this example, rather than preserving knowledge structures related to California, knowledge structures related to Vienna become more crucial. In summary, knowledge editing methods must determine the extent of generalization on specific edit samples.
>
> ---
>
> > **How do the knowledge editing methods compare to each other in terms of their effectiveness and efficiency, what is the takeaway in method selection?**
>
> 1. Generally, **the effectiveness of knowledge editing methods can be evaluated through the Reliability metric**. We only need to evaluate the model's performance on edited samples before and after editing. If the probability of predicting the edited target increases after editing, it signifies successful editing procedure. **Efficiency can be evaluated based on the consume for training, average time taken to edit a single sample, and inference time after editing.**
>
> 2. In selecting editing methods, we not only consider effectiveness and efficiency but also pay attention to factors like **Locality (whether it affects predictions on other unrelated samples)** and **the Robustness proposed in this paper (whether it introduces inconsistencies in knowledge and disrupts knowledge structures)**.
>
> ---
>
> > **In Multi-label Edit, how to guarantee the overall conceptual hierarchy among labels?**
>
> Apologies, we currently do not have a viable solution to maintain the conceptual hierarchy among labels. Achieving this likely requires training strategies supporting structured inputs, and there are challenges in evaluating this structurally. We hope to address this issue in future research.
>
> We hope these responses could address your questions. **Additionally, in the rebuttal revision, we will incorporate the minor issues as you proposed.** Once again, thank you for your comments and suggestions！

---

### Official Review · Reviewer_mRJP · 2023-10-31

**Soundness:** 4 excellent
**Presentation:** 3 good
**Contribution:** 4 excellent
**Rating:** 8
**Confidence:** 3

**Summary:**

This paper comprehensively explores the side effects of knowledge editing for large language models (LLMs), highlighting potential risks in real-world use cases. To facilitate a rigorous evaluation, the researchers introduce two innovative datasets specifically crafted to highlight the unintended consequences of knowledge editing. The study offers solutions for knowledge conflicts and introduces the MLE method to mitigate distortion risks. It also discusses implementation challenges and prospects of knowledge editing for LLMs.

**Strengths:**

The authors assess the risks associated with current knowledge editing methodologies for LLMs, and introduce two datasets for the purposes of finding potential drawbacks of LLMs.

This paper presents the MLE method as a straightforward solution to mitigate knowledge distortion risks and address potential knowledge conflicts.

The challenges and prospects of implementing knowledge editing for LLMs are discussed.

**Weaknesses:**

The paper's scope is limited to factual knowledge editing.

However, the presence or absence of knowledge conflicts or distortions in other types of knowledge editing remains unexplored.

The authors should supplement this part of the paper to make it more comprehensive.

**Questions:**

please refer to the weakness section.

---

> ### Author Response · Authors · 2023-11-19
> **Response to Reviewer mRJP**
>
> Thank you for taking the time to review our paper thoroughly and for providing valuable comments and feedback. In response to your concerns, below are the detailed answers.
>
> ---
>
> > **The paper's scope is limited to factual knowledge editing, However, the presence or absence of knowledge conflicts or distortions in other types of knowledge editing remains unexplored.**
>
> In our paper, we only consider specific types of knowledge editing techniques. For other known types of knowledge editing, our considerations are as follows:
>
> - **Unstructured Knowledge Editing:** Methods such as SERAC [1], MEND, and CaliNet [2] support unstructured knowledge editing, which can be achieved through QA-based model editing. The robustness issues of knowledge editing proposed in this paper depend on determining the editing scope and mining knowledge structure. In unstructured knowledge editing, for instance, “What is the highest mountain in the world? Mauna Kea → Mount Everest”, even if no (s, r, o)-formatted triplets can be extracted, traditional metrics used to evaluate individual knowledge edits (e.g., generalization and locality) are easily calculable. However, considering multiple edits as discussed in this paper, as most edit samples may lack any connection, **interference among multiple edits will be challenging to explore in unstructured knowledge editing**. Additionally, issues with the knowledge structure within the model also require exploration using accurate, structured knowledge bases like Wikidata to facilitate further research into the interpretability of knowledge editing and its reliable practical applications in LLMs.
>
> - **Common-sense Editing[3]:** There's limited research on the logical and reasoning chains of common-sense knowledge, **lacking a theoretical basis in defining the core problems explored in this paper** and constructing benchmarks.
>
> ---
>
> *[1] SERAC: Memory-based Model Editing at Scale (ICML 2022)*
>
> *[2] CaliNet: Calibrating Factual Knowledge in Pretrained Language Models (EMNLP 2022)*
>
> *[3] Editing Common Sense in Transformers (EMNLP 2023)*

---

### Author Response · Authors · 2023-11-19
**Summary of Revisions**

Dear Reviewers and AC:

We sincerely appreciate your valuable time and constructive comments.

We've uploaded a revised draft incorporating reviewer feedback. Modified texts are highlighted yellow. Below is a summary of the main changes:

- Add an illustration in Figure 1 to describe our work content.
- Add more complete evaluation in Knowledge Conflict, which is detailed in Appendix B.
- Add more details of Multi-Label Edit (MLE) in Appendix C.
- Fix a bug in the schematic diagram of Knowledge Conflict in the original Figure 1.
- Solve the overlap error of Coverage Edit in the origial Figure 3.
- Correct the typos mentioned by reviewers.

We sincerely hope our responses and revisions address all reviewers’ concerns.

We sincerely believe that these updates may help us better deliver the benefits of the proposed work to the ICLR community. Thank you very much,

Authors.

---

### Author Response · Authors · 2023-11-21
**Urgent Request for Re-review and Discussion**

Dear Reviewers and AC,

We deeply appreciate the thoughtful and constructive feedback you have provided regarding our manuscript. As the discussion period draws to a close on **November 22nd**, we kindly urge you to participate in the ongoing discussion and provide any additional insights or clarifications you may have. Your expertise is valuable to us, and we are confident that your further comments will significantly contribute to the improvement of our work.

Thank you immensely for your time and thoughtful consideration.  We look forward to hearing from you soon.


Sincerely,

All Authors

---

### Meta-Review · Program_Chairs · 2023-12-04

**Metareview:**

The paper proposes new evaluation settings for knowledge editing in LMs: knowledge conflict and knowledge distortion. The reviewers (and AC) appreciate new perspective and insight (mainly the limitations of the existing models) their evaluation provides.  The paper also proposes a simple mitigation strategy (multi label edit). The experiments are in general solid, covering many existing knowledge editing methods and two LMs.

minor suggestion: I think having multi-label edit in figure 2 is a bit confusing, as it is semantically very different (it’s a mitigation method) from all the other three items (which are problem definition?) I think it’d be better to formally define it even briefly in the section (new appendix helps!).

**Justification For Why Not Higher Score:**

I find related work relatively lacking. There has been multiple studies recently showing that knowledge editing methods are not robust, many are listed in related work section but not discussed carefully). Please strengthen this section for the final version. The problem scope is somewhat limited.

**Justification For Why Not Lower Score:**

The paper will be helpful for future work in this domain (knowledge editing).

---

### Decision · Program_Chairs · 2024-01-16

Accept (poster)